# Comparative Genomic Analysis and Metabolic Potential Profiling of a Novel Culinary-Medicinal Mushroom, *Hericium rajendrae* (Basidiomycota)

**DOI:** 10.3390/jof9101018

**Published:** 2023-10-15

**Authors:** Jing Wei, Min Cheng, Jian-fang Zhu, Yilin Zhang, Kun Cui, Xuejun Wang, Jianzhao Qi

**Affiliations:** 1Shangluo Key Research Laboratory of Standardized Planting & Quality Improvement of Bulk Chinese Medicinal Materials, College of Biology Pharmacy & Food Engineering, Shangluo University, Shangluo 726000, China; 2Shaanxi Key Laboratory of Natural Products & Chemical Biology, College of Chemistry & Pharmacy, Northwest A&F University, 3 Taicheng Road, Xianyang 712100, China; 3Qinba Mountains of Bio-Resource Collaborative Innovation Center of Southern Shaanxi Province, Hanzhong 723001, China

**Keywords:** *Hericium rajendrae*, comparative genome, cyathane diterpene, edible mushroom

## Abstract

*Hericium rajendrae* is an emerging species in the genus *Hericium* with few members. Despite being highly regarded due to its rarity, knowledge about *H. rajendrae* remains limited. In this study, we sequenced, de novo assembled, and annotated the complete genome of *H. rajendrae* NPCB A08, isolated from the Qinling Mountains in Shaanxi, China, using the Illumina NovaSeq and Nanopore PromethION technologies. Comparative genomic analysis revealed similarities and differences among the genomes of *H. rajendrae*, *H. erinaceus*, and *H. coralloides*. Phylogenomic analysis revealed the divergence time of the *Hericium* genus, while transposon analysis revealed evolutionary characteristics of the genus. Gene family variation reflected the expansion and contraction of orthologous genes among *Hericium* species. Based on genomic bioinformation, we identified the candidate genes associated with the mating system, carbohydrate-active enzymes, and secondary metabolite biosynthesis. Furthermore, metabolite profiling and comparative gene clusters analysis provided strong evidence for the biosynthetic pathway of erinacines in *H. rajendrae*. This work provides the genome of *H. rajendrae* for the first time, and enriches the genomic content of the genus *Hericium*. These findings also facilitate the application of *H. rajendrae* in complementary drug research and functional food manufacturing, advancing the field of pharmaceutical and functional food production involving *H. rajendrae*.

## 1. Introduction

Mushrooms are a collective term for the fruiting bodies of specific physical forms of fungi, which come in a wide variety of species. For centuries, people have been consuming mushrooms for their nutritional value and health benefits. Mushrooms play a significant role in agriculture, industry, and medicine, and their impact on economic development and environmental protection is significant and likely to increase in the future. Hericium, a genus of the Russulales order and Hericiaceae family, is a globally renowned medicinal and edible mushroom [1]. *Hericium erinaceus* is its most common and famous member, and it is not only a rare edible mushroom but also a well-known traditional Chinese medicine fungus with a long history of use. Currently, *H. erinaceus* has been proven to be a functional edible mushroom that could help prevent and delay various age-related neurological disorders [2]. Its chemical components exhibit extraordinary pharmacological effects, including neurotrophic [3], anti-aging, antioxidant, and anti-neuroinflammatory activities [4].

*Hericium rajendrae* is a relatively little known but valuable and rare species of the genus *Hericium*, with distinct morphological differences from *Hericium erinaceus*. Compared to *H. erinaceus*, which has been extensively studied for its chemical composition, little research has been conducted on the chemical composition of *H. rajendrae*. *Hericium alpestre,* another rare species within the genus *Hericium*, was reported to contain cyathane diterpenoids [5] and phenolic compounds [6,7], with the former found to stimulate the production of nerve growth factor and brain-derived neurotrophic factor [5], and the latter shown to have anticancer activity [6,7].

In recent years, the emergence of third-generation sequencing technology and the maturity of T2T assembly technology have made the gene sequencing and assembly of medicinal and edible fungi more accurate and efficient. These technologies promote research on various aspects such as fungal life cycles, mating types, nutritional modes, and biosynthesis of bioactive metabolites. With the support of efficient sequencing technology and mature assembly technology, the genome of some valuable medicinal fungi such as *Ganoderma lucidum* [8], mulberry Sanghuang [9], *H. erinaceus* [10], *Laetiporus sulphureus* [11] and *Inonotus hispidus* [12] have been successfully decoded. This research will further contribute to their medical use and industrial development. Furthermore, the genome sequencing of some precious wild edible fungi, including *Pleurotus giganteus* [13], *Agaricus sinodeliciosus* [14] and *A. bitorquis* [15], will also promote the strain breeding and artificial cultivation of these species.

The genus *Hericium* includes several species of medicinal and culinary value, such as *H. erinaceus*, *H. coralloides*, and *H. rajendrae*. While the genomes of *H. erinaceus* [10,16] and *H. coralloides* [17,18] have been extensively studied, the genome of *H. rajendrae* remains unexplored. Despite the availability of a published *H. rajendrae* genome, its poor quality did not meet the requirements of relevant studies. To fill this gap, we sequenced and analyzed the complete genome of a wild *H. rajendrae* strain. High-quality assembly and annotation showed that *H. rajendrae* is dikaryon, and contains fifteen pseudochromosomes and four contigs. This is the first time that the genome of *H. rajendrae* has been sequenced and analyzed. Comparative analysis with the genomes of *H. erinaceus* and *H. coralloides* provided new insights into the genome, mating system and carbohydrate metabolism capabilities of *H. rajendrae*. Considering the promising activity of *Hericium* mushrooms against neurodegenerative diseases, we further evaluated their potential for secondary metabolite biosynthesis, identified cyathane diterpenoids, and analyzed the biosynthesis of these compounds. This study adds to the diversity of *Hericium* genome research and provides new insights into the genetics and physiology of these rare medicinal and culinary mushrooms.

## 2. Materials and Methods

### 2.1. Fungal Strain and Strain Culture

The fruiting body from a natural habitat was collected from the side veins of Qinling Mountain in Zhashui County, Shaanxi Province, China. By comparing the morphological characteristics of the specimens and performing ITS sequence alignment of the mycelium (Appendix A), the sample was identified as *H. rajendrae* and subsequently classified as *H. rajendrae* NPCB A08. Tissue isolation was carried out using fresh wild fruiting bodies of the strain NPCB A08. In order to obtain cultivable mycelium, small pieces of sterilized basidiospores were placed on potato dextrose agar (PDA, Difco, BD, USA) plates and cultured for one week. The identified mycelium was preserved in the Key Laboratory of Natural Products and Chemical Biology, College of Chemistry and Pharmacy, Northwest A&F University. This process ensures the availability of viable mycelium for further studies.

### 2.2. Genome Sequencing, De Novo Assembly, and Annotation

#### 2.2.1. Extraction of Genome DNA

The cultivation of *H. rajendrae* NPCB A08 mycelium was carried out in a controlled environment using potato dextrose broth (PDB, Difco, BD, USA) medium at a temperature of 25 °C with agitation at 200 rpm for a duration of one week. The purpose of this cultivation was to obtain a sufficient amount of fresh and viable mycelium. To ensure the purity and freshness of the mycelium, a series of washing steps were performed, including centrifugation, followed by rinsing with sterile water and then re-centrifugation to remove excess water. Genomic DNA extraction from the mycelium was performed using the sodium dodecyl sulfate technique. This involved grinding the mycelium with liquid nitrogen, and the DNA quality and concentration were assessed by agarose gel (0.75%) electrophoresis, Nanodrop One spectrophotometer (Thermo Fisher Scientific, CA, USA) and Qubit 3.0 Fluorometer (Life Technologies, Carlsbad, CA, USA). Detailed information on the specific isolation and purification method can be found in a previous document [11].

#### 2.2.2. Sequencing and De Novo Assembly

After assessing the quality and integrity of the DNA, it was randomly fragmented using a Covaris ultrasonic disruptor (Covaris, Woburn, MA, USA). Subsequently, Illumina sequencing pair-end libraries were constructed using the Nextera DNA Flex Library Prep Kit (Illumina, San Diego, CA, USA) with an insert size of 300 bp. The sequencing process was carried out on the Illumina NovaSeq6000 platform (Illumina, San Diego, CA, USA). To ensure high-quality data, the raw reads were subjected to a cleaning process to remove low-quality reads. This was accomplished using the SOAPnuke v2.1.8 (https://github.com/BGI-flexlab/SOAPnuke, accessed on 12 May 2023). Following the data filtering step, the clean data were utilized for subsequent analyses. For Oxford Nanopore sequencing, the libraries were prepared using the SQK-LSK109 ligation kit (Oxford Nanopore Technologies, Oxford, UK) following the standard protocol. Subsequently, the purified library was loaded onto primed R9.4 Spot-On Flow Cells and subjected to sequencing using a PromethION sequencer (Oxford Nanopore Technologies, Oxford, UK). To obtain the sequence data from the raw reads, base-calling analysis was conducted using the Oxford Nanopore GUPPY v6.4.6 (https://community.nanoporetech.com/downloads, accessed on 10 May 2023).

The NECAT software (https://github.com/xiaochuanle/NECAT, accessed on 10 July 2023) was used for genome correction and assembly, resulting in an initial assembly. This was followed by two rounds of error correction using the Racon V1.5 with default parameters based on the third-generation sequencing data. Subsequently, two rounds of error correction with second-generation reads were performed using the Pilon 1.2.4 (https://github.com/broadinstitute/pilon, accessed on 12 July 2023) with default parameters. Finally, the corrected genome was further processed using purge_haplotigs to remove heterozygous sequences, resulting in the final assembled genome.

#### 2.2.3. Gene Prediction and Annotation

BRAKER v3.0.3 (https://github.com/Gaius-Augustus/BRAKER, accessed on 22 May 2023) was mainly used for gene sequence prediction. GeneMark—EX was then used to train the model, and AUGUSTUS v.3.5.0 (https://github.com/Gaius-Augustus/Augustus, accessed on 22 April 2023) was used for ORF prediction. The Rfam database was used to predict and classify non-coding RNAs using INFERNAL v1.1.4 (https://github.com/EddyRivasLab/infernal, viewed 22 April 2023). In addition, Repeat Modeller v2.0.4 (https://github.com/Dfam-consortium/RepeatModeler, accessed on 22 January 2023) was used to generate repetitive libraries by incorporating Rebase libraries, and Repeat-Masker v4.1.5 (https://github.com/rmhubley/RepeatMasker, accessed on 22 April 2023) was used for repetitive genomic sequence annotation. Finally, BLAST searches were performed against non-redundant protein sequences from National Center for Biotechnology Information (NCBI), Swiss-Prot, COG, and KEGG databases for gene product annotation. The above software runs on a dual-core server equipped with two Intel E5-2699V4 and 512 Gb RAM, running with default parameters.

### 2.3. Comparative Genomics Analysis

Genomic collinearity was analyzed and visualized using McscanX (https://github.com/wyp1125/MCScanX, accessed 4 July 2023). Clustering analysis of the comparative genomes of the genus *Hericium* species was performed using Orthofinder v2.5.5 (https://github.com/davidemms/OrthoFinder, accessed on 4 July 2023), which run using the following settings: -S diamond -M msa -T fasttree -t 88. The comparative genome results were visualized using jVenn (http://jvenn.toulouse.inra.fr/app/index.html, accessed 2 July 2023). To calculate the ratio of synonymous substitution rates (Ks) to nonsynonymous substitution rates (Ka) for each species of the genus *Hericium*, genome-wide replication analyses were performed using wgd v1.1.2 (https://github.com/arzwa/wgd, accessed 13 July 2023) and Para AT v2.0 (https://ngdc.cncb.ac.cn/tools/paraat, accessed 13 March 2023), and the results were visualized using Rstudio v4.20.

### 2.4. Transposon Element and the LTR-RT Analysis

Four types of transposon elements (TEs), including long terminal repeat (LTR), long interspersed element (LINE), short interspersed element (SINE) and DNA transposon element (DNA-TE), were predicted by RepeatModeler v2.0.2 (https://github.com/Dfam-consortium/RepeatModeler, accessed on 12 August 2023) and RepeatMasker v4.1.5 (https://www.repeatmasker.org/RepeatMasker/, accessed on 12 August 2023). First, the RepeatModeler software was used to construct a custom repeat library and then merge it with the repbase library. Repeat sequence annotation of the genome was then performed using RepeatMasker. LTR insertion times (LTR-RT) are identified and analyzed with the help of LTR_retriever v2.9.0 (https://github.com/oushujun/LTR_retriever, accessed on 12 August 2023).

### 2.5. Phylogenomic Analysis and Gene Family Variation Analysis

Phylogenetic analysis was performed to investigate the evolutionary relationships between *Hericium* strains and 37 other representative species of Basidiomycetes. Single-copy homologous genes were identified using OrthoFinder v2.5.5 with the parameters “-S diamond -M msa -T raxml-ng”. Divergence time prediction of 156 single-copy orthologous sequences from 40 strains was performed using the MCMCTree within PAML 4.9e (http://abacus.gene.ucl.ac.uk/software/paml.html, accessed on 12 March 2023). The calibrated points of several groups of recent ancestor divergence times were queried in TIMETREE 5 (http://www.timetree.org, accessed on 11 July 2023), including *Ganoderma sinense* vs. *Grifola frondosa* (84.2–135.2 MYA), and *Laetiporus sulphureus* vs. *Gelatoporia subvermispora* (137.2–164.3 MYA), *Lyophyllum decastes* vs. *Tricholoma matsutake* (90.6–118.1 MYA) and *Paxillus involutus* vs. *Suillus brevipes* (90.2–130.0 MYA). Gene family expansion and contraction were calculated using CAFÉ 4.2.1 (https://github.com/hahnlab/CAFE, accessed on 10 July 2023) with the identified orthologous gene families, which run as the following parameters: --cores 30 --fixed_lambda 0.0001.

### 2.6. CAZy Family, Microsatellite, and Cytochrome P450 Analyses

To annotate and classify genes encoding carbohydrate-active enzymes (CAZymes) from the genomes of three *Hericium* species and other edible fungi, the CAZy database (http://bcb.unl.edu/dbCAN2, accessed on 20 July 2023). A bubble plot of the CAZyme analysis was generated using the Complex Heatmap package in Rstudio v4.20.

Genome-wide microsatellites were identified using the Tandem Repeat Sequence Finder (TRF v4.09.1, https://github.com/Benson-Genomics-Lab/TRF, accessed on 20 July 2023) using default parameters. Simple sequence repeats (SSRs) with di- to hexa-nucleotide motifs were screened for marker development.

P450s were predicted and target protein sequences annotated using Diamond 2.1.8 (E-values < e^−5^) and the Hmmer v3.3.2 (filter parameter E-value < e^−5^; coverage > 0.35). The reference P450 sequences used for clustering analysis were obtained from the Fungal Cytochrome P450 Database (http://p450.riceblast.snu.ac.kr/index.php?a=view, accessed on 8 March 2022). For the phylogenetic tree analysis, 133 predicted P450 proteins from *H. rajendrae* NPCB A08 and several other Basidiomycota were selected from the fungal P450 database for clustering for accurate classification. Maximum likelihood trees were constructed using IQ-tree 2.2.2.6 (https://github.com/iqtree/iqtree2, accessed on 10 July 2023) with the options “-m MFP -bb 1000 -alrt 1000 -abayes -nt AUTO”.

### 2.7. Prediction and Cluster Analyses of Gene Clusters Involved in Secondary Metabolites

Prediction of biosynthetic gene clusters was accomplished with the fungal version of antiSMASH 7.0 (https://fungismash.secondarymetabolites.org/#!/start, accessed on 20 July 2023). Evolutionary tree-based cluster analysis is implemented via IQtree with the parameters described above. For a detailed analysis of multi-domain synthases such as NRPS and PKS, the package Synthaser [19] was used to analyze their domain characteristics. These structural domains include adenylation (A), acyl carrier protein (ACP), acyltransferase (AT), thiolation (T), thioesterase (TE), condensation (C), β-ketoacyl synthetase (KS), product template (PT), acyl carrier protein transacylase (SAT), thioesterase (TE), and thioester reductase (TR). The assessment of homology and similarity between two or more BGCs was performed using Clinker [20], which is based on the comparison of the sequence similarity of the encoded proteins. Visualization of the comparison results was achieved using clustermap.js [20], a tool embedded in Clinker to generate gene cluster comparison plots.

### 2.8. Metabolites Profiling of H. rajendrae NPCB A08

Five kg rice were divided into 100 1 L shake flasks and then 50 milliliters of sterile water was added. The mixture was soaked for two hours and then subjected to high pressure sterilization to prepare the culture medium (rice medium). After the strain was added to the culture medium, the culture was allowed to ferment at room temperature for 30 days. The culture was collected and subjected to three extractions with ten liters of ethyl acetate. The ethyl acetate extract was concentrated under reduced pressure, resulting in 76 g crude extract. Ten milligrams of the crude sample were subjected to high resolution mass spectrometry (HRMS) for GNPS (https://gnps.ucsd.edu) analysis, and the rest of the crude extract was separated by various chromatographic separation gears to obtain the monomer compounds. HRMS detection was performed using an AB Sciex TripleTOF 6600 mass spectrometer (AB Sciex, MA, USA) in positive ion modes. Molecular network analysis of HPLC-HRMS data of crude extracts was performed using GNPS (accessed on 17 July 2023) with default parameters. Finally, the molecular networks were visualized by Cytoscape 3.9.1. Details of the isolation of the monomeric compounds are described on the Appendix A. The structural characterization of the monomeric compounds was accomplished by high-resolution mass spectrometry and/or NMR spectroscopy, and the data were acquired via a Bruker Avance III 400 and 500 MHz NMR spectrometer, using TMS as an internal standard, with chemical shifts recorded in parts per million δ (ppm).

### 2.9. Data Availability

The ITS sequence of *H. rajendrae* NPCB A08 was registered in the NCBI GenBank under accession number OR646745 and the final genome assembly results and related data were submitted to NCBI under BioProject PRJNA1018320 and BioSample SAMN37432729, respectively. The network file based on positive-ion mode MS data can be found and accessed at https://gnps.ucsd.edu/ProteoSAFe/status.jsp?task=7fbc2f60d1954567b8e592475c2ae3e2 (accessed on 31 July 2023). The NMR data of compounds **1**, **4**, **5**, and **7** have been submitted to NP-MRD (Deposition ID: NPd000000335), and are available to download at this link: https://depositions.np-mrd,org/request-data/3424b45a-715d-4b7d-b57f-840e76057cf7 (accessed on 7 October 2023).

## 3. Results

### 3.1. Genome Sequence Assembly and Annotation of H. rajendrae NPCB A08

A pre-conducted genome survey based on Illumina sequencing indicated that the genome size of *H. rajendrae* NPCB A08 was less than 50 Mbp (Appendix A). The presence of two peaks with a two-fold relationship in the *K*-mer curve indicated that the genome of *H. rajendrae* NPCB A08 was heterozygous, with a heterozygosity of 3.61% (Figure 1A, Appendix A). This finding suggested that *H. rajendrae* was a dikaryon. The genome of *H. rajendrae* NPCB A08 was de novo assembled to 46.77 Mbp, which consists of fifteen pseudochromosomal molecules and four contigs, by combining Illumina sequencing data and nanopore sequencing data (Figure 1B, Table 1, and Appendix A). The completeness of the genome assembly was assessed by a coverage of 99 95% (Appendix A) and a BUSCO value of 91.6% (Appendix A) based on the fungi_odb10 database.

The assembly quality of *H. rajendrae* NPCB A08 was superior to that of *H. erinaceus* CS-4 and *H. coralloides* FP101451 in terms of contig number and N50 (Table 1). There were 13,622 genes predicted by BRAKER, equipped with Augustus, which contain 13,418 protein-coding genes (Appendix A), and 204 non-coding genes (Appendix A). The BUSCO evaluation showed that 84.2% of the 13,418 coding genes were single-copy genes. These genes had an average length of 1715.42 bp, consisting of 93,499 exons and 80,081 introns, with average lengths of 182.74 bp and 74.07 bp, respectively (Appendix A). In addition, 204 non-coding genes code 136 tRNAs, 38 rRNAs, 29 snRNAs, and one sRNA (Appendix A). These findings provide insight into the genome structure of *H. rajendrae*.

To achieve comprehensive functional annotation of protein-coding genes, we conducted sequence similarity analysis and motif similarity search on a dataset consisting of 13,418 genes. Public databases such as NCBI nr (Appendix A), GO (Appendix A), COG (Appendix A), Uniprot, KEGG Pathway (Appendix A), Pfam (Appendix A), Refseq, and Interproscan were utilized for this purpose. Our analysis successfully annotated 11,716 genes (87.31%) at least once (Appendix A). The annotation results obtained from the Nr library revealed that 9253 genes, accounting for 86.66% of all protein-coding genes, were annotated. Among these genes, 78.67% exhibited a significant match with *H. alpestre*, indicating a close relationship with Hericium species. Additionally, 10.77% showed similarity to *Dentipellis fragilis*, while 2.29% matched *Bondarzewia mesentenica* (Appendix A). This observation reflects the intergeneric variability of the *H. rajendrae* genome. These results emphasize the functional diversity of protein-coding genes in strain NPCB A08, as elucidated from various perspectives and levels of annotation.

### 3.2. Comparative Genomic Analysis within Hericium Species

Comparison of genome sizes indicates that the genome of *H. rajendrae* NPCB A08 is larger than that of *H. erinaceus* CS-4, but smaller than that of *H. coralloides* tvtc0002. However, the number of proteins encoded by *H. rajendrae* NPCB A08 is greater than that of both *H. erinaceus* CS-4, and *H. coralloides* tvtc0002 (Table 1). Collinearity analysis showed that the fifteen pseudochromosomes of *H. rajendrae* NPCB A08 have high homology with the fifteen chromosomes of *H. erinaceus* CS-4, but homologous relationships hardly exist on the same numbered chromosomes (Figure 1A). Further analysis showed that most genomic regions of these two *Hericium* species showed highly syntenic to that of *H. rajendrae* (Appendix A). Comparative analysis of orthologous genes among the three *Hericium* species identified a total of 7410 groups, meanwhile *H. rajendrae* NPCB A08 contains more unique orthologous groups (125) than that of *H. erinaceus* CS-4 (238) and *H. coralloides* tvtc0002 (86). The number of homologous genes shared between *H. rajendrae* NPCB A08 and *H. erinaceus* CS- 4 (1162) is much greater than that of *H. coralloides* tvtc0002 shared with *H. rajendrae* NPCB A08 (593) and *H. erinaceus* CS-4 (444), respectively (Figure 1C).

To gain further insight into these differences, a genome-wide duplication analysis based on nonsynonymous substitution rate (Ka) and synonymous substitution rates (Ks) were performed. The Ka/Ks curves of the three *Hericium* species were different. The trends of the Ka/Ks curves of *H. coralloides* and *H. erinaceus* were relatively close to each other, while the Ka/Ks curves of *H. rajendrae* were significantly different. This finding reflects that the three species have experienced different degrees of evolutionary selection pressure, while *H. rajendrae* has suffered more pure selection (Figure 1D).

Given the important role of transposon elements (TEs) in genomic constitute, TEs in the genus *Hericium* were used for further analysis. The content of four types of TE, LTR (Gypsy and Copia), LINE, SINE and DNA TEs, was investigated. The TE content in *H. rajendrae* (7.66% of the 46.77 Mb genome) was slightly higher than that identified in *H. coralloides* (6.05% of the 35.33 Mb genome) and *H. erinaceus* (5.76% of the 41.88 Mb genome), whereas it was much lower than that of *Lactarius deliciosus* 48 (23.70% of the 96.04 Mb genome), the control species (Figure 1E). Cross-genome comparisons showed that LTRs contributed the most to the TE expansion in the four taxa. *H. rajendrae* had 1462 Gypsy-LTRs, which were the dominant TEs and occupied 5.13% of the genome (Figure 1E). The continuous insertion of intact Gypsy- and Copia-LTRs in *H. rajendrae* since nearly one MYA was observed, and it had a distinct polymorphic distribution (Figure 1F). In the genera *Hericium* and *L. deliciosus*, the maximal bursts of LTRs were close to one MYA. However, a second burst of apparent LTR occurred in *H. erinaceus*, which occurred at approximately 16–17 MYA (Figure 1G).

### 3.3. Phylogenetic and Gene Family Variation Analysis

To shed light on the evolutionary history of H. rajendrae, a phylogenetic tree reconstruction and species divergence time estimation of 40 edible and medicinal fungi (Appendix A), with *Ustilago maydis* as an outgroup, were conducted from 156 conserved single-copy orthologous proteins (Figure 2). The inferred phylogenetic tree was strongly supported by bootstrap values. The mean divergence time of Russulales containing the genus *Hericium* and the group containing Ployporales, Atheliales, and Gloeophyllales was 172.115 MYA with 95% highest posterior density (HPD) of 112.559–231.349 MYA. The emergence of the genus Hericium is estimated to have occurred at a crown age of 27.503 MYA (95% HPD of 15.258–42.530 MYA). Among the three species of the genus *Hericium*, *H. rajendrae* has the closest evolutionary affinity with *H. erinaceus*, with a divergence time of 16.115 MYA (95% HPD of 8.737–25.586 MYA), whereas that between *H. rajendrae* and *H. coralloides* was estimated to be 204.64 Mya (95% HPD of 125.66–348.17 MYA).

Further analysis based on reconstructed evolutionary trees revealed complex gene contraction and expansion events in 28,144 gene families in the genomes of these 40 species (Appendix A). In the genus of *Hericium*, 108/42, 108/601, and 464/173 gene families were found to have undergone expansion/contraction in *H. coralloides*, *H. erinaceus,* and *H. rajendrae*, respectively. Among them, *H. rajendrae* had the most significant expansion events, affecting a total of 313 gene families (Figure 2).

### 3.4. CAZymes Analysis

CAZymes, as members of gene families, play a vital role in the genome. Specifically, CAZymes found in mushrooms have the ability to utilize cellulose- and lignin-rich substrates like wood chips and straw to obtain nutrients for their own growth and development [21,22]. Considering the nutritional value of *H. rajendrae* and its demand for large-scale cultivation, we conducted an investigation on its CAZymes. Our findings revealed the presence of 360 genes encoding 380 CAZymes in *H. rajendrae*. These included 186 glycoside hydrolases (GHs), 162 auxiliary activities (AAs), 60 glycosyltransferases (GTs), 35 glycoesterases (CEs), fourteen carbohydrate-binding modules (CBMs), and twelve polysaccharide lyases (PLs) (Figure 3A, Appendix A). Notably, twenty genes, such as g1089, encoded enzymes with dual CAZyme domains. Comparison of the CAZyme profiles of *H. rajendrae* with those of *H. coralloides* and *H. erinaceus* did not reveal any significant differences (Figure 3A, Appendix A). Furthermore, our analysis of 33 edible fungi’s CAZyme repertoires showed that the number and types of CAZymes were not species-specific. Among these fungi, the strains *Lactarius deliciosus* EDB83 exhibited the highest similarity to the genus *Hericium* in terms of their CAZyme profile (Figure 3A, Appendix A, Dataset S1). Additionally, *H. rajendrae*, *H. erinaceus*, and *H. coralloides* possessed twenty, ten, and fifteen CAZymes with dual domains, respectively. Cluster analysis indicated a high level of relatedness among most of the CAZymes with dual domains (Figure 3A,B).

### 3.5. Cytochrome P450 Family Analysis

Cytochrome P450s (CYPs) are another important family of genes in organisms that play important roles in various biological processes. In fungi, CYPs are extensively involved in their primary and stimulated metabolic processes. To gain a deeper understanding of the number and types of CYPs in *H. rajendrae*, we used Pfam prediction based on structural domain features to screen them. In total, we identified 121 genes encoding 133 different CYPs. Through clustering analysis, we determined a clear classification of the P450s in *H. rajendrae* NPCB A08 by comparing the protein sequences of these 133 proteins with representative fungal P450 sequences from the Cytochrome P450 Database (Appendix A). These clustering results clearly illustrate the classification of P450s in *H. rajendrae* NPCB A08. The cluster analysis revealed the presence of nineteen CYP subfamilies and eight uncertain groups (Figure 4). Among the identified CYP families, the CYP5144 family has the largest number of members with 30, followed by CYP5037 (20) and CYP5139 (10), while the remaining subfamily members are all less than 10. In addition, the 30 P450s are scattered in eight ambiguous groups, accounting for 22.56% (Figure 5). These unidentified P450s indicate the presence of potential novel P450 types that require further analysis and identification.

### 3.6. Identification of the Mating Genes and SSR Marker Development

The development of mushroom-shaped fruiting bodies in basidiomycetes is a complex process, in which mating is a key step regulated by specific mating loci [23]. Mating loci (MAT) are located in different chromosomal regions on the genome. Heterozygous cooperation dominates the mating types of fungi, which can be divided into a bipolar mating type and tetrapolar mating type. Among them, the tetrapolar mating system is the most extensive and complex sexual reproduction control system discovered so far in Basidiomycetes [24]. Considering that the reasons for the formation of the fruiting bodies of *Hericium* mushrooms are still unknown, and the demand for cultivation of the *H. rajendrae* due to the huge market vacancies, it is necessary to analyze and characterize its mating system.

The *matA* locus of *H. rajendrae* was found to be located on Chr1 by a homologous sequence search with the mitochondrial intermediate peptidase (*mip*, HE02G000648) codon gene and a homeodomain transcription factor 2-codon gene (*HD2*, HE02G000668) of *H. erinaceus* CS-4 [10]. Genes on the *matA* locus encode a glycosyltransferase family 8 protein (*glgen*, g5593), an unknown conserved fungal protein (*βfg*, g5596) two HD transcription factors (*HD1*, g5598, and *HD2*, g5621), an *mip* (g5599) (Figure 5A), whereas the *matB* locus contains at least five unclustered *ste3*, including g3950, g3952, g3953, g3965, and g3966, located on Chr2 (Figure 5B). The corresponding coding products of the *matA* and *matB* loci between *H. rajendrae* NPCB A08 and *H. erinaceus* CS-4 showed high similarity (Figure 5A,B). Current analyses indicate that the *matA* and *matB* loci are not located on the same chromosome, suggesting the existence of a quadrupolar mating system in *H. rajendrae*. However, this finding only reveals the intricate genomic structure of *H. rajendrae* mating loci. Further investigations are needed to fully understand sexual reproduction and fruiting body formation in *H. rajendrae*.

Microsatellite markers are an important molecular breeding tool that will have potential application in screening for elite varieties of *H. erinaceus* [10]. Therefore, microsatellite sequences in the genome of *H. rajendrae* were characterized and developed as genetic markers. Typical SSRs with repeat sequence lengths between one and six bp were searched on the genome of *H. rajendrae*, and a total of 1448 typical SSRs were found. The total length of SSRs was about 25.18 kb, accounting for 0.054% of the total genomic length. Among them, tri-nucleotide SSRs were the most abundant repeat type, accounting for 49.59% of the total repeats, followed by di-nucleotide (23.90%) and mono-nucleotide (17.75%) SSRs. Further comparisons showed that the SSR traits of *H. rajendrae* were highly similar to those reported for *H. erinaceus* [10] and predicted for *H. coralloides* (Figure 5C, Appendix A).

### 3.7. The BGCs for Secondary Metabolite Analysis

In view of the long edible history and significant medicinal value of *H. rajendrae*, we analyzed its biosynthetic potential of secondary metabolites. The genome of the strain NPCB A08 was predicted using the webtool antiSMASH, and 24 secondary metabolite biosynthetic gene clusters (BGCs) containing 31 core genes were discovered (Table 2). The core genes consisting of eleven terpenoid synthase, a non-ribosomal peptide synthase (NRPS) and 16 NRPS-likes enzymes, a polyketide synthase (PKS), and two ribosomally synthesized and post-translationally modified peptide like enzymes (RiPP-likes) are distributed on nine chromosomes (Chr2–6, 9–12, 14, and 15) (Figure 6A, Table 2).

Among the eleven terpenoid synthesis-related enzymes (Appendix A), there are eight sesquiterpene synthases (STSs). Cluster analysis of STSs from the strain NPCB A08 and identified STSs from the basidiomycete revealed eight STSs belonging to Clade I, II, and IV, and no members belonging to Clade III (1,11-cyclization of (2*E*, 6*E*)-FPP) (Figure 6B, Appendix A). Similarly, the FPPs of *H. erinaceus* and *H. coralloides* do not have members of Clade III. In the case of *H. rajendrae*, Clade II contained four FPPs, while Clade I and III each contain two FPPs (Figure 6B). Three enzymes associated with triterpene synthesis, squalene oxidase (g11649.t1), squalene synthase (g12602.t1), and lanosterol synthase (g12602.t1) were identified sequence similarity comparison (Appendix A).

Cluster analysis of the 16 NRPS-like enzymes revealed that g9771.t1 and g7538.t1 were in relative outgroup (Figure 6C). Further structural compositional analysis revealed that these two enzymes predicted by antiSMASH to be NRPS-like were labeled NRPS_PKS and NRPS, respectively, because the former contains a KS domain at the C-terminus and the latter contains a C domain at the N-terminus. The other 14 NRPS-like enzymes were uniformly characterized by A-T-TR domains (Figure 6D).

The PKS g84366.t1 shows almost 100% identify with HE04T003470.1, an identified orsellinic acid synthase (OAS) from *H. erinaceus* [25]. Hercor1_45461 from *H. coralloides* is another predicted OAS, which displays over 80.00% identify with g84366.t1 and HE04T003470.1. The identifies of these three OASs with other identified OASs of basidiomycetes ranged from 40.00% to 80.00% (Figure 6E). Domain analysis reflects that these OASs share almost the same domain composition (SAT-KS-AT-PT-ACP-ACP-TE), except for an additional ACP domain in PKS2 (Figure 6F). These OASs are speculated to synthesize OA using acetyl-coenzyme A and malonyl-coenzyme A as substrates (Figure 6G). Cluster 5 contains not only g8436.t1 and 2 NRPS-likes (g8434.t1 and g8441.t1), but also multiple postmodifying enzymes, which suggests that this BGC may encode structurally complex products (Figure 6G). Cluster 17 (g7538.t1) and cluster 20 (g9771.t1) each contain an unusual NRPS-like and several postmodifying enzymes, and it is worthwhile to investigate their products (Figure 6G).

### 3.8. Secondary Metabolic Profiling and Putative Biosynthesis for Erinacines

*H. rajendrae*, a rare species of the genus *Hericium*, has been poorly studied in terms of its metabolites. Considering the high genomic similarity among *H. rajendrae*, *H. erinaceus* and *H. coralloides*, the secondary metabolites of *H. rajendrae* were investigated by HPLC-HRMS with the help of GNPS (Appendix A). The structures of partial monomeric compounds were confirmed by NMR analysis. A total of fifteen compounds were identified by comparison of their MS and MS2 characteristics with the previous literatures, including erinacine A (**1**), erinacine B (**2**), erinacine G (**3**), erinacine E (**4**), erinacine F (**5**), erinacine T (**6**), erinacine Z1 (**7**), **8,** Hericinoid B (**9**), CJ-14,544 (**10**), CP-412,065 (**11**), and Hericinoid C (**12**) (Table 3, Figure 7 and S10). All of these compounds are cyathane-type diterpenoids. Among them, **1**, **4**, **5**, and **7** were further confirmed by NMR (Appendix A, Appendix A). Almost all of these compounds identified in *H. rajendrae* were found in *H. erinaceus* [2]. Compounds **1**, **2**, **4** and **5** have been reported in *H. flagellum*, a synonym specimen of *H. alpestre* [5].

The biosynthetic pathway for erinacines derived from *H. erinaceus* has been elucidated [32,33], and in view of the high similarity of the genomes of the three *Hericium* species and the convergence of the metabolites of *H. erinaceus* and *H. rajendrae*, it is speculated that the BGCs for erinacines are also present in the genomes of *H. rajendrae* and *H. coralloides*. Using the BGC for erinacines from *H. erinaceus* (yamabushitake Y2) as a clue [32], the BGCs for erinacines were identified from *H. rajendrae* and *H. coralloides*. The sequence identity of the protein encoded by the corresponding gene on the BGC is not less than 67% (Figure 8A). The enzymes encoded by g9677 and g9585 are responsible for the synthesis of geranylgeranyl diphosphate (GGPP) and the cyclization of GGPP to the cyathane diterpene backbone cyatha-3,12-diene. The three CPYs then sequentially hydroxylate C14 (g9586.t1), the methyl group of C12 (g9580.t1), and C11 to form cyathatriol. The C11 of cyathatriol then undergoes acylation to form 11-O-acetyl cyathatriol. 11-O-acetylcyathatriol is catalyzed by the rare xylosyltransferase (g9587.t1) to the cyathane xyloside, erinacine Q, which is then catalyzed by postmodifier proteins (g1840.t1 and g9579.t1) and non-enzymatic catalysts to form **1**, **2**, and possibly erinacine C (Figure 8B).

## 4. Discussion

The genus *Hericium* is a well-known taxon in the order Russulales because of its specialized morphology, wide range of biological activities, and valuable nutritional value. In this study, we report the genome of a wild rare *Hericium* species, *H. rajendrae*, from the Qinling Mountains, which is the first time that the genome of *H. rajendrae* is reported at the chromosome level. Due to recent advances in sequencing technology, the assembly and annotation quality of *H. rajendrae* is superior to those of previously reported *Hericium* species (Table 1). Genomic synteny (Figure 1B), LTR insertion time (Figure 1G), and CAZymes (Figure 3A) analysis revealed interspecific convergence among *Hericium* species, whereas comparative genomic analysis based on orthologous groups revealed that *H. rajendrae* contains more unique genes (Figure 1C) and Gypsy-LTRs (Figure 1F). Analysis of contraction and expansion based on single-copy immediate homologous genes showed that *H. rajendrae* possessed a much higher number of expansions than those of *H. erinaceus* and *H. coralloides* (Figure 2), which may be related to the presence of a large number of Gypsy-LTRs present in *H. rajendrae* (Figure 1E). These results provide valuable insights into the complexity of the genome of the genus *Hericium*.

Considering the rarity of the source of *H. rajendrae*, artificial domestication and subsequent large-scale cultivation are inevitable. Therefore, it is meaningful to analyze and characterize the mating system of *H. rajendrae*. SSR-based development of molecular breeding markers in a variety of edible mushrooms including *Agaricus bisporus* [34], *Lentinula edodes* [35], *Flammulina velutipes* [36], and *Pleurotus tuoliensis* [37] has gained in-depth and extensive research, which is an important contributor to the large-scale cultivation of edible mushrooms. SSR analysis of the genome of *H. rajendrae* shows diversity of repetitive sequences. The investigation of these repetitive sequences contributes to the development of molecular markers for the *H. rajendrae* genome.

Mushrooms exert certain well-known pharmacological effects by producing specific types of biologically active secondary metabolites. For example, ganoderic acids demonstrate immunomodulatory activity [38], and Styrylpyrone compounds derived from the genera *Inonotus* [39] and *Phellinus* [40] show antioxidant properties. The anti-neurodegenerative activity of *H. erinaceus* is closely associated with the production of erinacines [3] and hericenones [2]. While the biosynthesis for erinacines has been identified in *Hericium erinaceus* [32,33], this study suggests that their BGCs also exist in *H. rajendrae* and *H. coralloides* (Figure 8). Given the promising medicinal value of erinacines, the identification of the BGCs for erinacines in *H. rajendrae* and *H. coralloides* is more significant. Hericenones are a class of OA-containing meroterpenoids, and the OAS from *Hericium erinaceus* have been uncovered [25]. Similarly, the OAS in *H. rajendrae* and *H. coralloides* were also identified in this study (Figure 6E,F). Considering the scarcity of sources of these compounds, the reconstruction of their biosynthetic pathways using *Aspergillus oryzae* to achieve high production [25,32] shows remarkable application value.

## 5. Conclusions

*Hericium rajendrae* is a rare medicinal edible mushroom with multiple biological activities and nutritional value. In this study, we present, for the first time, the complete genome of *H. rajendrae* at the chromosomal level. Comparative genomic analysis revealed the sequence similarities and compositional differences in the genome among *Hericium* species, and phylogenetic and evolutionary analysis revealed the divergence time of the *Hericium* genus and the variations in their gene families. The well-assembled genome and functional annotation provided important clues for the study of mating loci, CAZymes, and SSRs in this wild mushroom, contributing to its artificial cultivation and elite strain breeding. Cluster analysis reflected the diversity of CYPs in *H. rajendrae* and the convergence of FPPs in the Hericium genus. Core genes-based investigations reflected the biosynthetic diversity of secondary metabolites in *Hericium* species. Metabolite profiling based on molecular networks and gene cluster comparisons revealed the BGC for erinacines in *H. rajendrae* and inferred the biosynthetic pathway of erinacines in *H. rajendrae*. This study not only provides a high-quality *H. rajendrae* genome for the first time, but also delves into genomic characteristics and functional elements, including mating genes and biosynthesis core genes. These findings enrich the genomic content of the *Hericium* genus and will contribute to the development and utilization of *H. rajendrae* as alternative medicine agents and functional foods.

## Figures and Tables

**Figure 1 jof-09-01018-f001:**
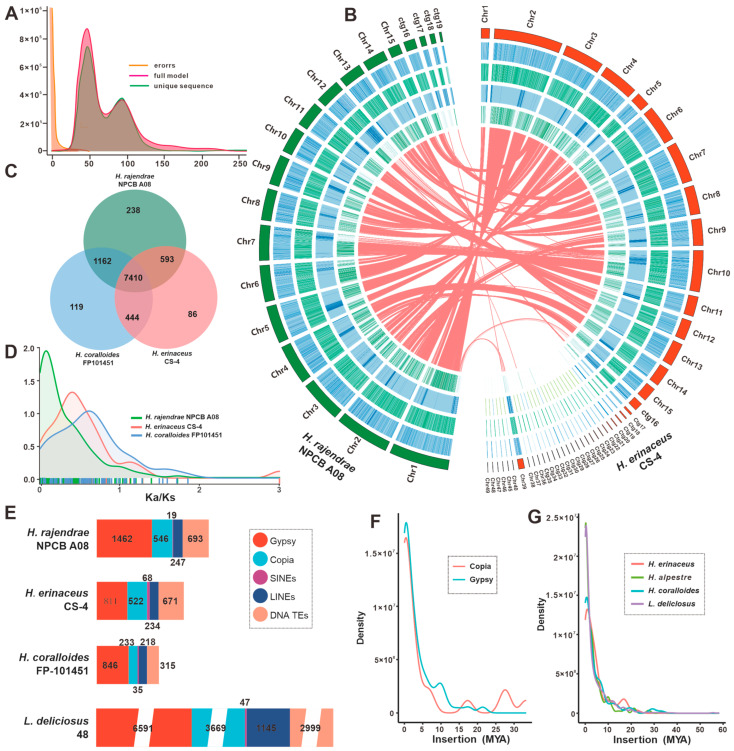
Genomic characterization and comparative genomic analysis. (**A**) Genomic collinearity analysis between *H. rajendrae* NPCB A08 and *H. erinaceus* CS-4. From the outside to the inside are I. Chromosome and Contigs; II–IV. GC-density, GC-skew, AT-skew (window size 10 kb), V. Gene-density (window size 100 kb), VI. Whole-genome collinearity analysis based on protein-coding genes between *H. rajendrae* NPCB A08 and *H. erinaceus* CS-4. (**B**) K-mer assessment curves for genome size and heterozygosity. (**C**) Venn schematic of comparative genomes within the genus *Hericium.* (**D**) Ka/Ks comparison within the genus *Hericium*. (**E**) Comparison of TE families in the genus *Hericium* and *L. deliciosus*. MYA indicates million years ago. (**F**) Insertion bursts of Gypsy and Copia elements in *H. rajendrae* NPCB A08. (**G**) Comparison of temporal patterns of intact LTR insertion bursts in the four taxa.

**Figure 2 jof-09-01018-f002:**
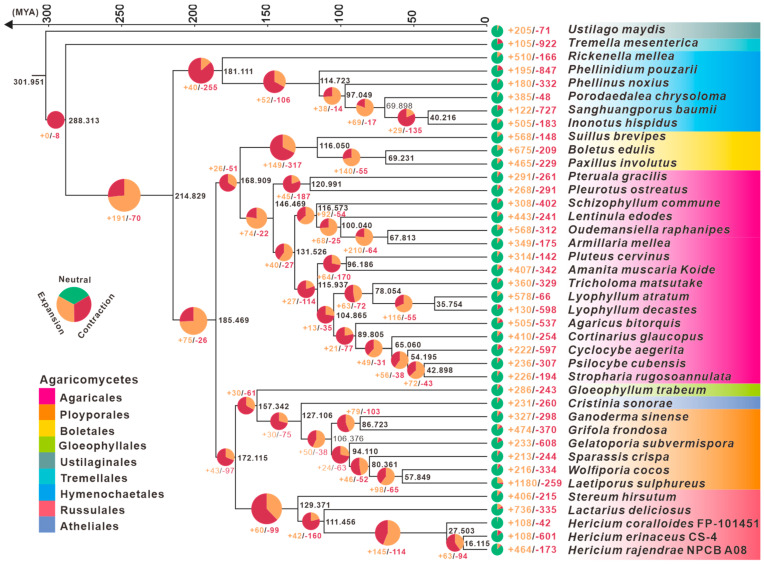
Phylogeny and gene family variation. The evolutionary relationship and expanded and contracted gene families among *Hericium* species and 37 representative medicinal Basidiomycetes. The maximum likelihood method credibility tree was inferred from 40 single-copy orthologous genes. All nodes received full bootstrap support. The divergence time is labeled as the mean crown age for each node, while the 95% highest posterior density is also given within the *Hericium* clade. The black numbers at the branches indicate the corresponding divergence times in MYA. The proportion of expansion and contraction in the genome of each species was displayed before its species name.

**Figure 3 jof-09-01018-f003:**
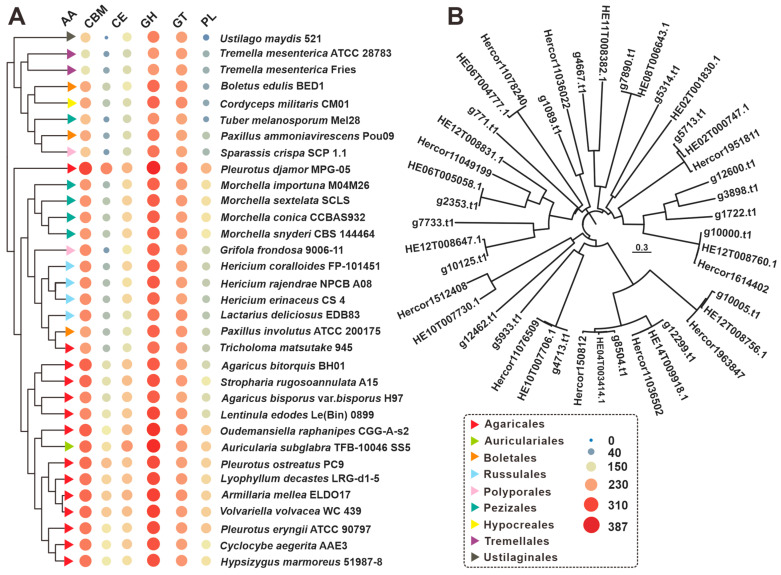
CAZymes analysis of *Hericium* and related edible mushrooms. (**A**) Composition comparison of CAZymes among 33 edible fungi including *H. rajendrae* NPCB A08. (**B**) Clustering analysis of CAZymes with bi-domains of *Hericium* mushrooms.

**Figure 4 jof-09-01018-f004:**
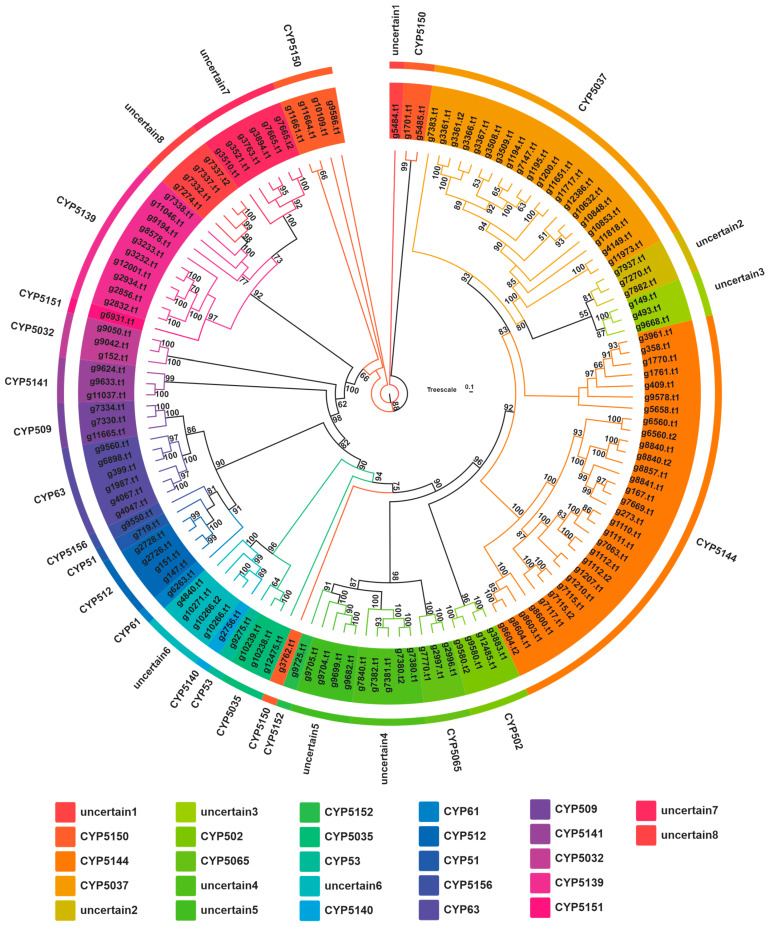
Maximum likelihood tree of 133 cytochrome P450s from *H. rajendrae* NPCB A08. Each cytochrome P450 family is shown in a separate color, and the branch reliability value of over 50 is marked on the corresponding branch node.

**Figure 5 jof-09-01018-f005:**
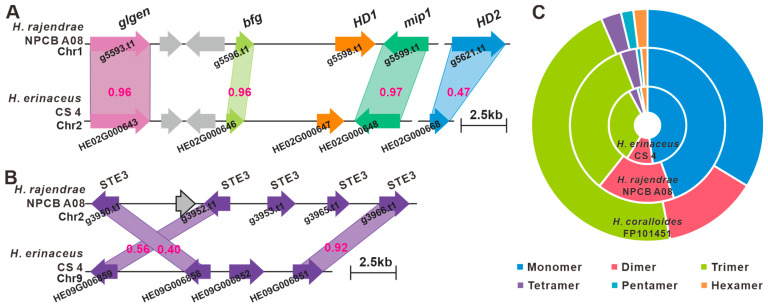
Identification of the mating genes and comparison of SSR abundance of *Hericium* species. Structural diagram of the genes on the *matA* locus (**A**) and *matB* locus (**B**) of *H. rajendrae*, the numbers on the similarity diagrams indicate the identity between corresponding genes. (**C**) Relative abundance of six type SSRs in the genus *Hericium*.

**Figure 6 jof-09-01018-f006:**
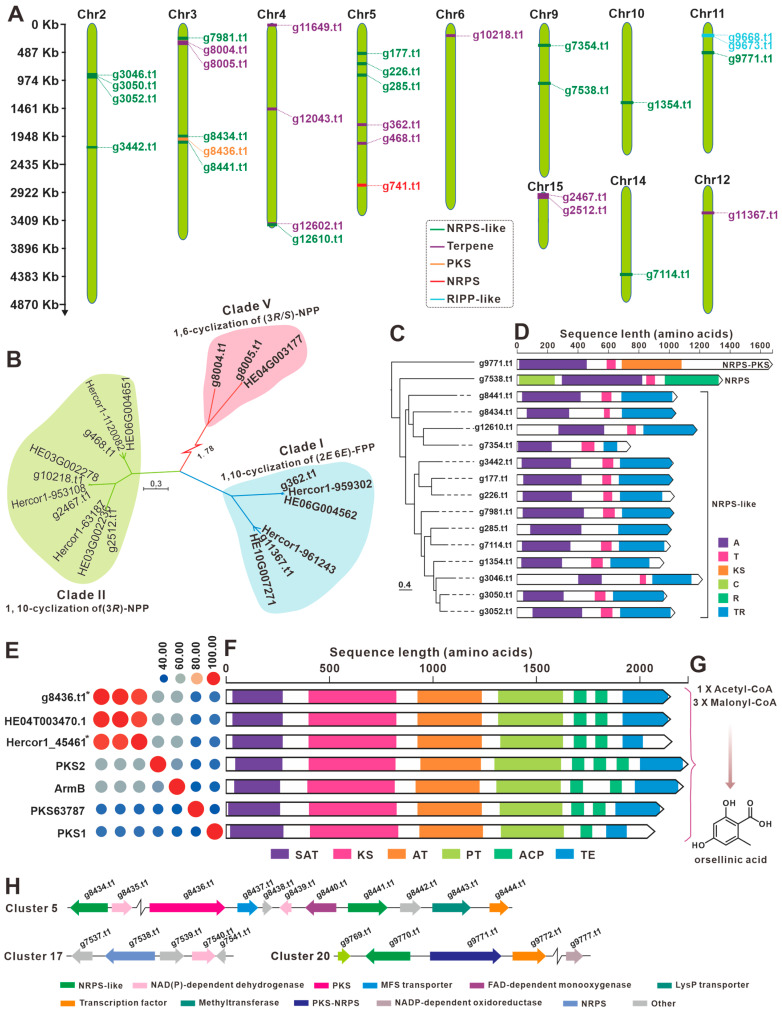
Analysis of genes involved in secondary-metabolite biosynthesis. (**A**) Distribution of biosynthetic core genes for natural products on the chromosomes, (**B**,**D**) phylogenetic tree analysis for STSs (**B**) and NRPS-likes (**D**), (**C**,**F**) domain analysis of NRPS-likes (**C**) and OASs (**F**), (**E**) percent identity matrix of seven OASs of Basidiomycota, an asterisk indicates that the function of the corresponding gene is predicted, (**G**) schematic representation of OA biosynthesis, and (**H**) schematic diagram of the composition of postulated clusters 5, 17, and 20.

**Figure 7 jof-09-01018-f007:**
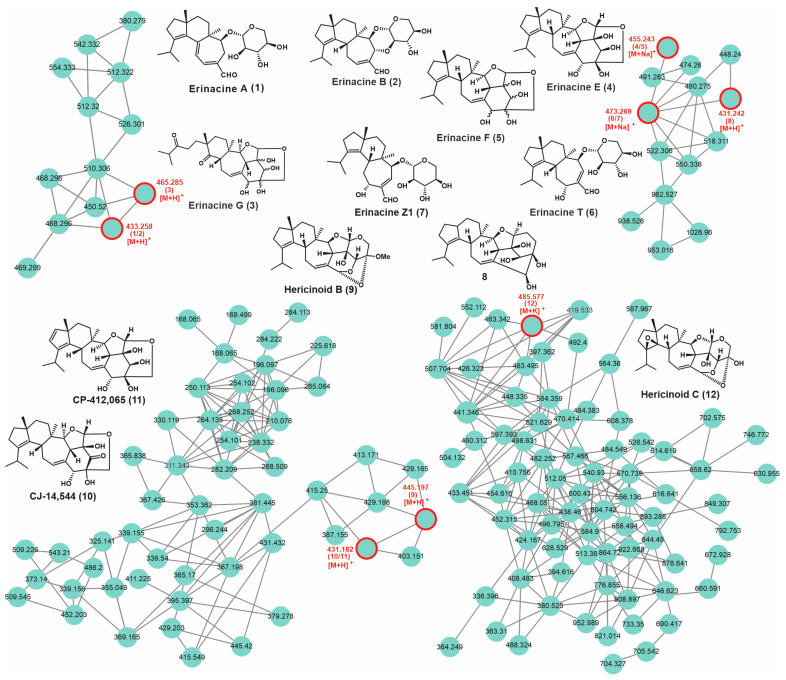
GNPS-based molecular network identification of metabolites from the fruiting bodies of *H. rajendrae*.

**Figure 8 jof-09-01018-f008:**
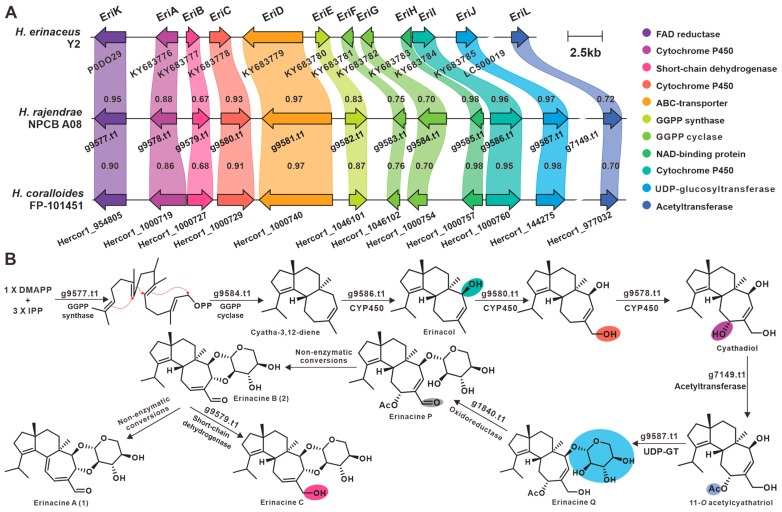
Biosynthesis of erinacines from the genus *Hericium*. (**A**) Comparison of BGCs for erinacines from the three *Hericium* species. (**B**) Proposed biosynthetic pathway of erinacines from *H. rajendrae*. GGPP synthase indicates geranylgeranyl diphosphate synthase, GGPP cyclase indicates geranylgeranyl diphosphate cyclase. UDP-GT indicates glycosyltransferase.

**Table 1 jof-09-01018-t001:** Genomic comparison within *Hericium* species.

Entry	*H. rajendrae* NPCB A08	*H. erinaceus* CS-4	*H. coralloides* FP101451
Sequencing technology	Illumina NovaSeq 6000 Nanopore PromethION	PacBio RSII, Illumina Hiseq X-Ten	PacBio
Sequencing depth	118×	750×	122×
No. of contig	19	52	125
No. of chromosome	15	15	NA
Total length (bp)	46,767,965	41,880,340	55,905,675
Largest length (bp)	5,307,752	6,077,030	1,366,710
Contig N50 (bp)	3,238,877	3,208,415	711,881
BUSCO (%)	91.6	96.4	97.5
Heterozygosity (%)	3.61	0.0226	0.847
GC content (%)	52.57	52.30	53.64
No. of protein-coding genes	13,418	10,620	12,369
GenBank accession No.	PRJNA1018320	GCA_006506795.2	JGI_Hercor1
References	This study	[10]	[18]

NA indicates not available.

**Table 2 jof-09-01018-t002:** Putative BGCs responsible for secondary metabolites in the strain NPCB A08.

Cluster No.	Location	Start (bp)	End (bp)	Core Gene ID	Core Gene Type
1	Chr2	860,561	963,214	g3046.t1	NRPS-like
				g3050.t1	NRPS-like
				g3052.t1	NRPS-like
2	Chr2	2,123,606	2,187,764	g3442.t1	NRPS-like
3	Chr3	228,685	291,856	g7981.t1	NRPS-like
4	Chr3	318,507	352,028	g8004.t1	terpene
				g8005.t1	terpene
5	Chr3	1,992,954	2,074,528	g8434.t1	NRPS-like
				g8436.t1	T1PKS
				g8441.t1	NRPS-like
6	Chr4	26,215	52,908	g11649.t1	terpene
7	Chr4	1,486,121	1,513,187	g12043.t1	terpene
8	Chr4	3,476,834	3,542,535	g12602.t1	terpene
				g12610.t1	NRPS-like
9	Chr5	493,923	554,595	g177.t1	NRPS-like
10	Chr5	674,234	733,369	g226.t1	NRPS-like
11	Chr5	868,813	933,044	g285.t1	NRPS-like
12	Chr5	1,747,264	1,775,280	g362.t1	terpene
13	Chr5	2,071,028	2,102,333	g468.t1	terpene
14	Chr5	2,804,202	2,824,688	g741.t1	NRPS
15	Chr6	198,427	229,557	g10218.t1	terpene
16	Chr9	354,078	414,897	g7354.t1	NRPS-like
17	Chr9	1,012,150	1,076,675	g7538.t1	NRPS-like
18	Chr10	1,369,643	1,433,801	g1354.t1	NRPS-like
19	Chr11	197,542	257,730	g9668.t1	fungal-RiPP
				g9673.t1	fungal-RiPP
20	Chr11	492,774	559,259	g9771.t1	NRPS-like
21	Chr12	460,518	492,887	g11367.t1	terpene
22	Chr14	1,503,013	1,567,208	g7114.t1	NRPS-like
23	Chr15	2	11,721	g2467.t1	terpene
24	Chr15	115,836	146,976	g2512.t1	terpene

**Table 3 jof-09-01018-t003:** The identified secondary metabolites from *H. rajendrae* NPCB A08.

No.	Putative Metabolite	Molecular Formula	Adduct	*m/z*	Reference
**1**	Erinacine A	C_25_H_36_O_6_	[M + H]^+^	433.258	[26]
**2**	Erinacine B	C_25_H_36_O_6_	[M + H]^+^	433.258	[26]
**3**	Erinacine G	C_25_H_36_O_8_	[M + H]^+^	465.285	[27]
**4**	Erinacine E	C_25_H_36_O_6_	[M + Na]^+^	455.243	[27]
**5**	Erinacine F	C_25_H_36_O_6_	[M + Na]^+^	455.243	[27]
**6**	Erinacines T	C_25_H_38_O_7_	[M + Na]^+^	473.269	[28]
**7**	Erinacine Z1	C_25_H_38_O_7_	[M + Na]^+^	473.269	[5]
**8**	**8**	C_26_H_38_O_5_	[M + H]^+^	431.242	[29]
**9**	Hericinoid B	C_26_H_36_O_6_	[M + H]^+^	445.197	[30]
**10**	CJ-14,544	C_25_H_34_O_6_	[M + H]^+^	431.182	[31]
**11**	CP-412,065	C_25_H_34_O_6_	[M + H]^+^	431.182	[30]
**12**	Hericinoid C	C_25_H_34_O_7_	[M + K]^+^	485.577	[30]

## Data Availability

Not applicable.

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
