# Peer review of "Comparative Genomic Analysis and Metabolic Potential Profiling of a Novel Culinary-Medicinal Mushroom, Hericium rajendrae (Basidiomycota)"

_jof, 2023, doi:10.3390/jof9101018_

Round 1

Reviewer 1 Report

The authors analyze a medical mushroom, Hericium rajendrae, in terms of comparative genomics and metabolomics. There are many careless mistakes in this manuscript. The authors should carefully check the manuscript again.

After minor revisions, the paper would be acceptable for publication in Journal of Fungi.

Minor points

Line 143: “2.3.” might be “2.4”.

Line 201: The authors should mention the relationship between “the culture medium” and PDA described in section 2.1.

Line 229: “50 M” should be replaced by “50 Mbp”.

Line 233: “46.77 Mb” should be replaced by “46.77 Mbp”.

Line 254: “H. alpestre” should be italicized.

Line 309: “H. rajendrae” should be italicized.

Line 311: “Ustilago maydis” should be italicized.

Lines 314-319: Both “MYA” and “Mya” are used. If they have the same meaning, one of these should be used.

Line 392: Figure 6A might have to be Figure 5A.

Line 394: Figure 6B might have to be Figure 5B.

Line 396: 6A-6B might have to be 5A-5B.

Line 417: “H. rajendrae” should be italicized.

Line 430: “Clade III” should not be italicized.

Line 442: “is” should not be italicized.

Line 449: g84366.t1 might have to be g8436.t1.

Line 451: “contain” might have to be replaced by “contains”.

The manuscript is well written, but there are some careless mistakes.

Author Response

The authors analyze a medical mushroom, Hericium rajendrae, in terms of comparative genomics and metabolomics. There are many careless mistakes in this manuscript. The authors should carefully check the manuscript again.

After minor revisions, the paper would be acceptable for publication in Journal of Fungi.

A:  Thanks for all of your time and effort (and to the referees), which we believe has resulted in a significantly improved manuscript.

Minor points Line 143: It seems that "2.3." should actually be "2.4".

A1: Thank you for your thorough review. The necessary changes have been made accordingly. Line 201: The authors should clarify the relationship between "the culture medium" and PDA as mentioned in section 2.1.

A2: Thank you for your careful attention. The appropriate changes have been made to address this concern.

Line 229: "50 M" should be replaced with "50 Mbp".

A3: We appreciate your keen observation. The necessary changes have been made accordingly.

Line 233: "46.77 Mb" should be replaced with "46.77 Mbp".

A4: Thank you for pointing that out. The correction has been made to reflect the accurate information.

Line 254: "H. alpestre" should be italicized.

A5: Thank you for your meticulous review. The required formatting adjustment has been made.

Line 309: "H. rajendrae" should be italicized.

A6: We appreciate your attention to detail. The necessary italicization has been applied.

Line 311: "Ustilago maydis" should be italicized.

A7: Thank you for bringing that to our attention. The appropriate formatting has been applied.

Lines 314-319: Both "MYA" and "Mya" are used. If they have the same meaning, please use only one of them.

A8: Thank you for your thorough review. The duplicate reference has been addressed by using a consistent abbreviation for simplicity.

Line 392: Figure 6A may need to be corrected to Figure 5A.

A9: Thank you for catching that mistake. The figure reference has been updated accordingly.

Line 394: Figure 6B may need to be corrected to Figure 5B.

A10: We appreciate your careful review. The figure reference has been corrected as suggested.

Line 396: 6A-6B may need to be corrected to 5A-5B.

A11: Thank you for noting that inconsistency. The figure references have been adjusted accordingly.

Line 417: "H. rajendrae" should be italicized.

A12: Thank you for your thoroughness. The necessary formatting adjustment has been made.

Line 430: "Clade III" should not be italicized.

A13: We appreciate your attention to detail. The formatting has been revised accordingly.

Line 442: "is" should not be italicized.

A14: Thank you for bringing that to our attention. The formatting has been corrected.

Line 449: g84366.t1 may need to be corrected to g8436.t1.

A15: Thank you for pointing out the error. The correct transcription has been applied.

Line 451: "contain" may need to be replaced with "contains".

A16: We appreciate your careful review. The necessary change has been made.

Reviewer 2 Report

This study aimed to build up the scientific basis for the better utilization of H. rajendrae. The whole genome sequence and annotation filled in the gap of the field. Comparative genomic analysis also tried to reveal the evolutionary characteristics of the species. Analysis on secondary metabolism and biosynthesis pathways is also interesting.

However, the MS is not mature yet and major revision is required to be published, especially that many critical information was missing for the assessment of “Results”. In addition, authors may consider combining Results and Discussion.

Introduction

1.      Karyotype and chromosome numbers, etc. of H. rajendrae should be introduced.

Materials and Methods

2.      Section 2.1: Redo the screen capture of Figure S1. The current highlighted row misleads the audience.

3.      Section 2.2.2: Reagents, protocol, sequencing chip, etc. for the generation of ONT and Illumina sequencing data were not recorded. ONT sequencing chemistry and sequencing mode must be addressed. Software and setting for ONT basecalling must be provided.

4.      Section 2.2.3-2.4: Parameters/cut-offs must be recorded for the in silico analyses.

5.      Section 2.5: What kinds of sequences were supplied to OrthoFinder, protein, representative protein? How was the filtering performed? Gene family variation analysis? This information was not found.

6.      Section 2.6: Diamond 2.1.8 (e-values > e-5)?

7.      Section 2.8: Chromatographic separation/NMR conditions must be described.

8.      Section 2.9: Sequencing raw reads should be distributed to public databases, NGDC/NCBI/etc.

Results

9.      Section 3.1 Line 231: heterozygosity. With the combination of ONT and Illumina, the authors might consider performing the haplotype phasing and comparisons to provide new insights, for example, biased haplotype usage and regulation.

10.  Section 3.1 Line 233: “which consists of 14 fifteen pseudochromosomal molecules”. Which number is correct?

11.  Section 3.1 Table S3: The sequencing yield did not match the general throughput of PromthION chips. Did multiplexing performed during the library preparation? Why 20 kb insert size is chosen? Did centromere and telomere sequences being identified?

12.  Section 3.1 Line 236: It seems that Table S4 cannot tell “a coverage of 99.95%”.

13.  Section 3.1 Table 1: Chromosome numbers of the species should be included. Contig N50 of JGI_Hercor1 should be assessable with the whole genome sequence fasta. In addition, the audience will be happy to have BUSCO values of all three species listed, evaluated with the same database.

14.  Section 3.1 Table S6: Number of genes with COG annotations is much lower than expected, even when compared to “JGI_Hercor1”. Did the authors try using other annotation approaches and databases?

15.  Section 3.1 Figure S2: Please modify the illustration.

16.  Section 3.3 Figure 2: Parameters of gene family expansion/contraction analysis were not found in the MS and the number here seems weird. Did any cut-off be applied, such as the significance of gene family undergone expansion/contraction and the number of genes within the gene family? In addition, the genome assembly quality/de novo transcriptome assembly quality also affects the gene family analysis even though only gene/cds/protein sequences are used in the analysis. Fragmented genome assemblies/incomplete transcriptome will mislead the gene family evolutionary analysis. A table must be supplied to list out the source of data used here (Hericium species and 39 representitive Basidiomycetes), with the summary on genome/transcriptome assembly quality. More information should be included before further assessment.

17.  Section 3.3 Figure 2: any gene families showed expansion/contraction with functional importance? Any results that could be linked with the section 3.4 and afterwards?

18.  Section 3.4 Figure 3A: How was the tree being constructed/species being clustered?

19.  Section 3.4 Figure 3B: Hard to tell the which sequence belongs to which species.

20.  Section 3.5 Line 367: Figure number.

21.  Section 3.6: Identification of mating type gene was not described in Materials and Methods.

22.  Section 3.7: What’s the meaning of t1 in g*.t1 for this section? Transcript 1?

23.  Section 3.7 Table S11: Any conserved/function domain checked? Low identity value of g8004, g8005, g12602.

NA

Author Response

Comments and Suggestions for Authors

This study aimed to build up the scientific basis for the better utilization of H. rajendrae. The whole genome sequence and annotation filled in the gap of the field. Comparative genomic analysis also tried to reveal the evolutionary characteristics of the species. Analysis on secondary metabolism and biosynthesis pathways is also interesting. However, the MS is not mature yet and major revision is required to be published, especially that many critical information was missing for the assessment of “Results”. In addition, authors may consider combining Results and Discussion.

A: We would like to express our gratitude to the reviewer for taking the time and effort to review our manuscript, as well as for providing valuable suggestions on its content. We greatly appreciate the reviewer's patience and professionalism, and we admire their careful examination of our manuscript. Their review comments and suggestions have played a significant role in improving our writing and enhancing the content of our manuscript. This review is not only a recognition of our efforts but also a guidance for our improvement. Through their guidance, we have gained a clearer understanding of the shortcomings in our manuscript and can further refine and elevate its quality. We will carefully consider and address every suggestion and comment provided by the reviewer.

Introduction

  1. Karyotype and chromosome numbers, etc. of H. rajendrae should be introduced.

A1: Thanks, we have made adjustments as you suggestion.

Materials and Methods

  1. Section 2.1: Redo the screen capture of Figure S1. The current highlighted row misleads the audience.

A2: Thanks for the careful review, modified as suggestion.

  1. Section 2.2.2: Reagents, protocol, sequencing chip, etc. for the generation of ONT and Illumina sequencing data were not recorded. ONT sequencing chemistry and sequencing mode must be addressed. Software and setting for ONT basecalling must be provided.

A3: Sections 2.2.1 (part) and 2.2.2 have been rewritten as you requested. Relevant detailed information has been disclosed in detail.

  1. Section 2.2.3-2.4: Parameters/cut-offs must be recorded for the in silico analyses.

A4: The relevant necessary information has been added.

  1. Section 2.5: What kinds of sequences were supplied to OrthoFinder, protein, representative protein? How was the filtering performed? Gene family variation analysis? This information was not found.

A5: The input file for constructing species evolutionary trees based on orthologous single-copy genes using OrthoFinder defaults to the total protein information for a single species, and its run parameters have been added to the revised manuscript.

  1. Section 2.6: Diamond 2.1.8 (e-values > e-5)?

A6: Corrected.

  1. Section 2.8: Chromatographic separation/NMR conditions must be described.

A7: Added.

  1. Section 2.9: Sequencing raw reads should be distributed to public databases, NGDC/NCBI/etc.

A8: The raw data has been uploaded to NCBI SAR databases and accessible links have been added to the revised manuscript, and the corresponding information is described in Section 2.8.

Results

  1. Section 3.1 Line 231: heterozygosity. With the combination of ONT and Illumina, the authors might consider performing the haplotype phasing and comparisons to provide new insights, for example, biased haplotype usage and regulation.

A9: We thank the reviewers for their constructive suggestion, which will be adopted in the subsequent manuscripts.

  1. Section 3.1 Line 233: “which consists of 14 fifteen pseudochromosomal molecules”. Which number is correct?

A10: Corrected.

  1. Section 3.1 Table S3: The sequencing yield did not match the general throughput of PromthION chips. Did multiplexing performed during the library preparation? Why 20 kb insert size is chosen? Did centromere and telomere sequences being identified?

A11: Because sequencing can be stopped when the expected amount of data is reached, the amount of data did not reach the theoretical maximum amount that can be produced by one chip.

The 20 kb insert size is not chosen because if most of the fragments are found to be smaller than 20 kb after DNA extraction, then we will choose to re-extract them. There is no filtering of fragment length in the library building and sequencing process.

The identification and analysis of telomeres and filaments have received little attention in the analysis of fungal (mushroom) genomes and were not investigated in this study.

  1. Section 3.1 Line 236: It seems that Table S4 cannot tell “a coverage of 99.95%”.

A12: Thanks, change has been made.

  1. Section 3.1 Table 1: Chromosome numbers of the species should be included. Contig N50 of JGI_Hercor1 should be assessable with the whole genome sequence fasta. In addition, the audience will be happy to have BUSCO values of all three species listed, evaluated with the same database.

A13: Thank you for your constructive comments on Section 3.1. We calculate Contig N50 values and have added information about chromosome numbers and Contig N50. In addition, we evaluated the BUSCO values for the three species using fungi_odb10 and have updated the table accordingly.

  1. Section 3.1 Table S6: Number of genes with COG annotations is much lower than expected, even when compared to “JGI_Hercor1”. Did the authors try using other annotation approaches and databases?

A14:  Annotating the genome using 9 databases, of which COG has the least annotated results, is normal and indisputable. There is no such thing as an expected value here. In addition, among the annotation results of the genome of H. coralloides FP101451 (JGI_Hercor1), the existence of COG's annotation results is unknown, and the proportion of COG's annotation results is even more unknowable. Annotating genomes using nine databases is a recognized strategy for genome annotation, and no annotation method other than these was used in this study.

  1. Section 3.1 Figure S2: Please modify the illustration.

a15: Modified.

  1. Section 3.3 Figure 2: Parameters of gene family expansion/contraction analysis were not found in the MS and the number here seems weird. Did any cut-off be applied, such as the significance of gene family undergone expansion/contraction and the number of genes within the gene family? In addition, the genome assembly quality/de novo transcriptome assembly quality also affects the gene family analysis even though only gene/cds/protein sequences are used in the analysis. Fragmented genome assemblies/incomplete transcriptome will mislead the gene family evolutionary analysis. A table must be supplied to list out the source of data used here (Hericium species and 39 representitive Basidiomycetes), with the summary on genome/transcriptome assembly quality. More information should be included before further assessment.

A16: The methods for gene family variation analysis have been added in Section 2.5, and key parameters are described. Since genome quality (assembly quality and annotation results) may affect gene family variation analysis, 37 genomes of medicinal mushrooms with high quality and genomes of the genus Hericium mushroom were selected for analysis. The genome sources of the species used for gene family variation analysis were summarized and presented in Table S9. In fact, H. coralloides FP101451 may not have good genome assembly quality (125 contigs), but species evolution based on single-copy orthologous proteins construction still clearly clusters the three species of the genus Hericium on a minimal branch. Furthermore, the number of gene family contractions and expansions in H. coralloides does not differ significantly from the other two Hericium species.

  1. Section 3.3 Figure 2: any gene families showed expansion/contraction with functional importance? Any results that could be linked with the section 3.4 and afterwards?

A18: Gene family expansion/contraction is used to characterize the variation in the content of the genome of a particular species during its evolution, and is a commonly used method of genome analysis. The results of this analysis are informatively described and summarized in Section 3.3.

  1. Section 3.4 Figure 3A: How was the tree being constructed/species being clustered?

A18: This tree is essentially a representation of the clustering relationships used to characterize the quantitative relationships between species in the six categories of CAZy Family. The construction method has been shown in the first paragraph of Section 2.6.

  1. Section 3.4 Figure 3B: Hard to tell the which sequence belongs to which species.

A19: The entries used for clustering are the names of their corresponding protein sequences, which can be found in their genome annotation files.

  1. Section 3.5 Line 367: Figure number.

A20: Corrected.

  1. Section 3.6: Identification of mating type gene was not described in Materials and Methods.

A21: Identification of mating type gene is essentially the process of sequence alignment, which is described along with the results in the second paragraph of Section 3.5 due to its oversimplification.

  1. Section 3.7: What’s the meaning of t1 in g*.t1 for this section? Transcript 1?

A22: These mean different transcripts and are also used in specific contexts to refer to their corresponding amino acid sequences.

  1. Section 3.7 Table S11: Any conserved/function domain checked? Low identity value of g8004, g8005, g12602.

A23: These proteins were annotated through 9 databases, some of which were annotated with libraries investigating their conserved/function domains. This table just shows the results of these proteins related to terpene biosynthesis by comparing them with the UniportKB database.

As for g8004, g8005, g12602, the low identity values are perfectly acceptable, and some proteins with possible new functions or structures should be allowed to exist. After all, our knowledge of the mushroom genome is still shallow.

Reviewer 3 Report

The manuscript entitled "Comparative genomic analysis and metabolic potential profiling of a novel culinary-medicinal mushroom, Hericium rajen
drae (Basidiomycota)" represents a detailed genomic characterization of a previous unsequenced species of fungi. The authors present also a metabolomics analysis for the potent characterization of interesting secondary metabolites. 

The detailed work of the authors is of great merit and sets high standards for future fungi genomic characterizations. Nevertheless, there are some details that authors need to correct for considering publication to the prestigious journal of fungi. 

Minor comments

Section 2.2.1. Extraction of Genome DNA

Please write at least briefly the procedure of DNA extraction. In my experience isolating DNA from fungi can be sometimes challenging therefore at least a brief protocol should be presented. Just a reference from the previous method is not sufficient.

Sequencing and De Novo Assembly

I am a little troubled about the description of the assembly method. I cannot understand if the authors first assembled Illumina reads and utilized them for error correctness of nanopore sequencing after separately assembling nanopore reads. Please spend some time to be more descriptive about the software used for short reads assembly and long reads assembly for better corresponding the quality result to a quality method description.

Family names, gene names, genera and species should be italicized throughout the manuscript.

199 just Five kg

233 de novo not de nova

The authors use a number of hardware and reagents in which they do not present the company name and city. Please fix that according to journal guidelines.

No comments

Author Response

The manuscript entitled "Comparative genomic analysis and metabolic potential profiling of a novel culinary-medicinal mushroom, Hericium rajendrae (Basidiomycota)" represents a detailed genomic characterization of a previous unsequenced species of fungi. The authors present also a metabolomics analysis for the potent characterization of interesting secondary metabolites. The detailed work of the authors is of great merit and sets high standards for future fungi genomic characterizations. Nevertheless, there are some details that authors need to correct for considering publication to the prestigious journal of fungi. 

A:Thank you for acknowledging and carefully reviewing our work. We appreciate your positive evaluation and attention to detail. Thank you again for your valuable input, which will undoubtedly strengthen the manuscript.

Minor comments

Section 2.2.1. Extraction of Genome DNA

Please write at least briefly the procedure of DNA extraction. In my experience isolating DNA from fungi can be sometimes challenging therefore at least a brief protocol should be presented. Just a reference from the previous method is not sufficient.

A: Thank you for your feedback and suggestions. We appreciate your expertise and agree that providing a more detailed procedure for DNA extraction is important, especially considering that isolating DNA from fungi can be challenging at times. In response to your comment, we will include a brief protocol outlining the steps involved in the DNA extraction process in Section 2.2.1. This addition will provide clarity and ensure that readers have a clear understanding of our methodology.

Sequencing and De Novo Assembly

I am a little troubled about the description of the assembly method. I cannot understand if the authors first assembled Illumina reads and utilized them for error correctness of nanopore sequencing after separately assembling nanopore reads. Please spend some time to be more descriptive about the software used for short reads assembly and long reads assembly for better corresponding the quality result to a quality method description.

A: Thank you for your comment regarding the assembly method described in our manuscript. We apologize for any confusion caused by our description. We understand the importance of providing a detailed explanation of the software used for both short reads assembly and long reads assembly. In our revised version, we will provide a more comprehensive description of the software used for each step, highlighting how they contribute to the overall quality of the assembly.

Thank you for bringing this to our attention, and we appreciate your feedback.

Family names, gene names, genera and species should be italicized throughout the manuscript.

A: Thank you for your meticulous review. The required formatting adjustment has been made.

199 just Five kg

A: Thanks, modified as suggestion.

 233 de novo not de nova

A: Thank you, the correction has been made as suggested

The authors use a number of hardware and reagents in which they do not present the company name and city. Please fix that according to journal guidelines.

A: Thank you for bringing up this concern regarding the hardware and reagents used in our study. In our revised version, we will ensure that the company names and cities associated with the hardware and reagents used are included in the appropriate sections of the manuscript. This will help to provide transparency and facilitate reproducibility of our work. Thank you for pointing out this oversight, and we appreciate your attention to detail.

Round 2

Reviewer 2 Report

I appreciate the authors' effort and the MS is much improved. However, minor corrections are still required.

1. Karyotype and chromosome number should be introduced in H rajendrae biology at the very beginning of the introduction.

2. Line 180: CAFE 4.2.1 and data visualization. Whether filtering by p value is applied?

3. Section 2.9: Please double check if the sequencing raw data is uploaded.

4. Line 340: Figure number?

Moderate editing is required.

Author Response

Comments and Suggestions for Authors

I appreciate the authors' effort and the MS is much improved. However, minor corrections are still required.

Thank you for recognizing the reworked draft, and the corrections or modifications you pointed out have been made.

  1. Karyotype and chromosome number should be introduced in H rajendrae biology at the very beginning of the introduction.

A1: Modified.

  1. Line 180: CAFE 4.2.1 and data visualization. Whether filtering by p value is applied?

A2: A default p-value was used for filtering, i.e. p=0.01.

  1. Section 2.9: Please double check if the sequencing raw data is uploaded.

A3: The relevant data had begun to be submitted on September 18, but NCBI was slow to process them. The progress is as follows:

The screenshot below is a screenshot of the email to submit the raw sequencing data.

  1. Line 340: Figure number?

A4: Revised